# Effect of Selected Antidepressants on Placental Homeostasis of Serotonin: Maternal and Fetal Perspectives

**DOI:** 10.3390/pharmaceutics13081306

**Published:** 2021-08-20

**Authors:** Hana Horackova, Rona Karahoda, Lukas Cerveny, Veronika Vachalova, Ronja Ebner, Cilia Abad, Frantisek Staud

**Affiliations:** Department of Pharmacology and Toxicology, Faculty of Pharmacy in Hradec Kralove, Charles University, Akademika Heyrovskeho 1203, 500 05 Hradec Kralove, Czech Republic; horackha@faf.cuni.cz (H.H.); karahodr@faf.cuni.cz (R.K.); cervenyl@faf.cuni.cz (L.C.); vachalovav@faf.cuni.cz (V.V.); ronjatebner@gmail.com (R.E.); abadmarc@faf.cuni.cz (C.A.)

**Keywords:** antidepressants, placenta, pregnancy, transport, serotonin, fetal programming

## Abstract

Depression is a prevalent condition affecting up to 20% of pregnant women. Hence, more than 10% are prescribed antidepressant drugs, mainly serotonin reuptake inhibitors (SSRIs) and selective serotonin and noradrenaline reuptake inhibitors (SNRIs). We hypothesize that antidepressants disturb serotonin homeostasis in the fetoplacental unit by inhibiting serotonin transporter (SERT) and organic cation transporter 3 (OCT3) in the maternal- and fetal-facing placental membranes, respectively. Paroxetine, citalopram, fluoxetine, fluvoxamine, sertraline, and venlafaxine were tested in situ (rat term placenta perfusion) and ex vivo (uptake studies in membrane vesicles isolated from healthy human term placenta). All tested antidepressants significantly inhibited SERT- and OCT3-mediated serotonin uptake in a dose-dependent manner. Calculated half-maximal inhibitory concentrations (IC_50_) were in the range of therapeutic plasma concentrations. Using in vitro and in situ models, we further showed that the placental efflux transporters did not compromise mother-to-fetus transport of antidepressants. Collectively, we suggest that antidepressants have the potential to affect serotonin levels in the placenta or fetus when administered at therapeutic doses. Interestingly, the effect of antidepressants on serotonin homeostasis in rat placenta was sex dependent. As accurate fetal programming requires optimal serotonin levels in the fetoplacental unit throughout gestation, inhibition of SERT-/OCT3-mediated serotonin uptake may help explain the poor outcomes of antidepressant use in pregnancy.

## 1. Introduction

Depression is one of the most frequent complications during pregnancy. Estimates of its prevalence vary significantly, but recent data indicate that up to 20% of pregnant women experience depression and/or anxiety [1]. Moreover, epidemiological studies have detected negative effects of untreated depression during gestation on pregnancy outcomes [2]. Thus, currently up to 13% of pregnant women are prescribed one or more antidepressants, most commonly selective serotonin reuptake inhibitors (SSRIs) and serotonin/noradrenaline reuptake inhibitors (SNRIs) [3,4]. These drugs inhibit reuptake of serotonin (5-hydroxytryptamine, 5-HT) in neuronal cells by blocking its specific transporters in the presynaptic membrane, thus increasing extra-neuronal 5-HT concentrations in the brain. However, SSRIs/SNRIs can also cross the placental barrier and affect 5-HT homeostasis in the fetoplacental unit, thus compromising placental functions and fetal development [5,6]. Therefore, use of antidepressants by pregnant women remains controversial.

5-HT is a crucial monoamine neuromodulator and developmental signaling molecule in the central nervous system. It regulates neuronal cell proliferation, migration, brain wiring, and provides proper brain development. Alterations in fetal brain 5-HT signaling have been associated with various negative neurological outcomes [7]. It is also important for the development and function of several other organs and body systems, including the heart, lungs, gastrointestinal tract, and hypothalamic–pituitary-axis [8]. Thus, strictly controlled levels of 5-HT in the fetoplacental unit throughout gestation are crucial for maintaining ideal in utero conditions for fetal development and optimal programming [9]. Consequently, any insult to homeostatic mechanisms of 5-HT during gestation may increase risks of poor pregnancy outcomes. Numerous epidemiological studies have indicated a link between the use of antidepressants in pregnancy and negative effects on fetal development and programming. These include, inter alia, autism spectrum disorder, schizophrenia, depression, attention-deficit/hyperactivity disorder [10,11,12,13,14,15,16,17] disturbance of the hypothalamic-pituitary axis [5,18], preeclampsia [19,20], metabolic and cardiovascular diseases [21], and pulmonary hypertension [22]. Interestingly, male offspring have higher susceptibility to autism and developmental delays after exposure to SSRIs than female offspring, indicating that fetal sex influences risks for adverse effects of prenatal antidepressants [23]. Additionally, while direct teratogenic effects of antidepressants are disputed, increased risks of lung, heart, and neuronal malformations have been described in the literature [24,25,26,27]. Lastly, newborns exposed to antidepressants in utero are reported to suffer from lower birth weight, preterm delivery, lower Apgar score, and withdrawal syndrome [28,29,30,31]. However, the mechanistic causes are still not fully elucidated.

The placenta, despite being a non-neuronal tissue, has similar 5-HT homeostasis machinery to that of the brain. In detail, placental trophoblast cells express 5-HT transporter (SERT/SLC6A4) in the apical, mother-facing membrane and organic cation transporter 3 in the basal, fetus-facing membrane (OCT3/SLC22A3). Additionally, trophoblast cells are fully equipped with 5-HT-producing and 5-HT-degrading enzymes: tryptophan hydroxylase (TPH) and monoamine oxidase A (MAO-A), respectively [32]. The placenta is thus capable of regulating 5-HT levels in the fetoplacental unit in accordance with placental and fetal demands, which are related to gestational age in both humans [33] and rats [34]. Specifically, at the beginning of pregnancy, the embryo/fetus lacks the capacity to synthesize 5-HT and is fully dependent on maternal or placental sources [35]. Towards term, the fetus can synthesize 5-HT from maternally derived tryptophan [36,37] and placental synthesis of 5-HT declines [33]. Importantly, term placenta shows increased expression of SERT, OCT3, and MAO-A, the three most important components of a “5-HT-detoxification mechanism”, in the fetoplacental unit [33,34].

We have recently shown that both human and rat term placenta extract 5-HT from the fetal circulation via OCT3, a poly-specific, high-capacity transporter [32]. We also demonstrated that placental OCT3 can be inhibited by various compounds of endogenous (e.g., glucocorticoids) and exogenous (e.g., pharmaceuticals) origin [32]. The potential capacity of selected antidepressants to inhibit OCT3 function has been recently reported [38], but the relevance of this interaction in the placental barrier has not been previously investigated. We thus hypothesized that antidepressants may interfere with prenatal 5-HT homeostasis by affecting its placental clearance on both maternal and fetal sides of the placenta, thus resulting in suboptimal 5-HT concentrations in the fetoplacental unit. To test this hypothesis, we used membrane vesicles isolated from human term placentas and evaluated effects of frequently used SSRIs (paroxetine, citalopram, fluoxetine, fluvoxamine, sertraline) and an SNRI (venlafaxine) on 5-HT uptake from both maternal and fetal circulations. Using in situ perfused rat term placenta, we further evaluated these drugs’ effects on placental extraction of 5-HT from fetal circulation, and the potential contributory role of fetal sex.

The placenta also expresses ATP-binding cassette (ABC) transporters, including three located in the apical membrane: P-glycoprotein (P-gp, ABCB1), breast cancer protein (BCRP, ABCG2), and multidrug resistance-associated protein (MRP2, ABCC2). These three transporters protect the fetus against maternal pharmacotherapy, but current literature does not provide unequivocal evidence about antidepressants’ interaction with placental efflux transporters [39]. Thus, as ABC transporters may contribute to reductions of antidepressants’ concentrations in the fetal circulation, we also investigated interactions of selected antidepressants with P-gp, BCRP, and MRP2 using MDCK-transfected cells and in situ perfused rat term placenta.

## 2. Materials and Methods

An outline of the research design and experimental approaches used in this study is shown in Figure 1.

### 2.1. Materials

[^3^H]5-hydroxytryptamine ([^3^H]5-HT), 80 Ci/mmol; [^3^H]dihydroalprenolol, levo-[propyl-1,2,3-3H] hydrochloride, 80 Ci/mmol; [^3^H]paroxetine 20 Ci/mmol; [^3^H]sertraline, 80 Ci/mmol, and [^3^H]citalopram, 70 Ci/mmol were purchased from American Radiolabeled Chemicals, Inc. (St. Louis, MO, USA). Unlabeled 5-HT hydrochloride, paroxetine hydrochloride, citalopram hydrobromide, sertraline hydrochloride, fluoxetine hydrochloride, fluvoxamine maleate, and venlafaxine hydrochloride were purchased from Sigma-Aldrich (St. Louis, MO, USA). Bicinchoninic acid assay (BCA assay) reagents were purchased from Thermo Fisher Scientific (Rockford, MA, USA) and Tri Reagent solution was obtained from the Molecular Research Centre (Cincinnati, OH, USA). All other chemicals were of analytical grade.

### 2.2. Preparation of Microvillous and Basal Membrane Vesicles from Human Term Placenta

Human term placentas (*n* ≥ 4) were obtained from uncomplicated pregnancies immediately after delivery at the University Hospital in Hradec Kralove.

Maternal decidua and chorionic plate were removed, and placental villous tissue was cut into small pieces. The tissue was homogenized in a solution containing 250 mM sucrose, 10 mM Tris-Hepes (pH 7.2), 5 mM EGTA, 5 mM EDTA and 1 mM phenylmethylsulfonyl fluoride (PMSF). Microvillous and basal membrane (MVM and BM, respectively) vesicles were prepared simultaneously, using a previously described differential centrifugation method based on Mg^2+^ precipitation and separation of MVM and BM in a sucrose gradient [40]. All these procedures were performed at 4 °C.

MVM and BM samples were resuspended in intravesicular buffer (290 mM sucrose, 5 mM Hepes, 5 mM Tris; pH 7.4), and vesiculated by passaging 20 times through a 25-gauge needle as previously described [32]. Protein concentration was measured using the BCA assay. The purity and enrichment of the membrane fractions were also determined as previously described [32] and are presented in Appendix A.

#### Concentration-Dependent Inhibition by Antidepressants of 5-HT Uptake by MVM and BM Vesicles

Uptake of [^3^H]5-HT into MVM or BM vesicles was measured at room temperature using the rapid vacuum filtration technique as previously described [32]. Briefly, MVM or BM vesicle (*n* ≥ 4) suspension was preincubated for 10 min in the presence of each of the antidepressants (separately), then uptake was initiated by adding 100 nM [^3^H]5-HT with the selected antidepressant. Effects of the drugs at six concentrations (0.1 nM, 1 nM, 10 nM, 100 nM, 10 µM, and 100 µM), relative to uptake by antidepressant-free controls, were tested in these experiments. Finally, the reaction was stopped by addition of ice-cold stop solution (130 mM NaCl, 10 mM Na_2_HPO_4_, 4.2 mM KCl, 1.2 mM MgSO_4_, 0.75 mM CaCl_2_; pH 7.4) and filtration through a 0.45 µM mixed cellulose ester filter (MF-Millipore, HAWP00010) under vacuum. Filter-associated radioactivity was determined by liquid scintillation counting. Non-specific tracer binding to the filter and plasma membranes was accounted for by subtracting radioactivity measured in protein-free controls and uptake at time zero, respectively, from the total vesicle uptake measurements.

### 2.3. In Situ Perfusion of Rat Term Placenta

Placenta perfusion experiments were performed with female Wistar rats (weight 350–450 g) purchased from Velaz, Ltd. (Prague, Czech Republic) as previously described [41,42]. Rats were anesthetized with pentobarbital in a dose of 40 mg/kg administrated into the tail vein. Term placentas were used in two perfusion modes: open-circuit perfusion of the fetal side to investigate effects of antidepressants on placental uptake of 5-HT from fetal circulation (one male and one female placenta was perfused from each dam), and dually perfused placenta in closed setup to evaluate transport of antidepressants across the placenta (one placenta was perfused from each dam).

#### 2.3.1. Effect of Antidepressants on Placental Extraction of 5-HT from Fetal Circulation; Influence of Fetal Sex

1 nM [^3^H]5-HT was added to the fetal reservoir with the selected antidepressants (each tested separately at 1 and 100 µM). Placental venous effluent on the fetal side of each preparation (and controls with no antidepressant) was collected for 40 min and the placental extraction ratio (ER) of 5-HT was calculated as ER = (C_fa_ − C_fv_)/C_fa_ × 100 where C_fa_ is the 5-HT concentration in the fetal reservoir entering the perfused placenta via the umbilical artery and C_fv_ is the 5-HT concentration in the umbilical vein effluent [32]. Fetal sex was assessed by measuring the anogenital distance and confirmed by genotyping as described previously [32]. For each condition, 4 pregnant dams were used and from each one male and one female placenta was perfused.

#### 2.3.2. Effect of Efflux Transporters on Placental Transport of Antidepressants

In the closed circuit (recirculation) setup, both maternal and fetal sides of the placenta were perfused (at flow rates of 1 and 0.5 mL/min, respectively) with identical concentrations (1 μM) of selected antidepressants: [^3^H]paroxetine, [^3^H]citalopram, or [^3^H]sertraline and the fetal perfusate was recirculated. More than 3 pregnant dams were used for each condition. Samples were collected at 10 min intervals from the maternal and fetal reservoirs and the fetal/maternal concentration ratio at equilibrium was calculated [41]. Subsequently, the effect of GF 120918 (2 µM), an inhibitor of P-gp and BCRP, was evaluated.

At the end of all perfusion experiments, the placenta was further perfused for 10 min with radioactivity-free Krebs buffer and collected. A fraction of placental tissue was snap-frozen for further PCR and Western Blot analyses, while the rest was weighed and dissolved in Solvable tissue solubilizer (PerkinElmer Life and Analytical Sciences, Boston, MA, USA) to detect tissue-bound radioactivity.

### 2.4. RNA Isolation from Rat Placental Tissue

Total RNA was isolated from perfused rat placenta samples using Tri Reagent solution following the manufacturer’s instructions. The purity of the isolated RNA was checked by measuring the A260/A280 ratio, and the integrity of the RNA samples was confirmed by electrophoresis on a 1.5% agarose gel. Total RNA concentrations were calculated from A260 measurements.

### 2.5. Reverse Transcription and Droplet Digital PCR Assay

1 µg of total RNA was reverse transcribed to cDNA in 20 µL reaction mixtures using an iScript Advanced cDNA Synthesis Kit (Bio-Rad, Hercules, CA, USA) and a Bio-Rad T100^TM^ Thermal Cycler (Hercules, CA, USA) according to the manufacturers’ instructions.

Absolute quantification of *Slc22a3* transcripts in rat placenta (*n* ≥ 18) was performed by duplex droplet digital PCR (ddPCR) analysis, as previously described [32]. Briefly, each reaction mixture consisted of 10 µL of ddPCR™ Supermix for Probes, 1 µL of TaqMan^®^
*Slc22a3* probe (FAM; Rn00570264_m1, Thermo Fisher Scientific, Waltham, MA, USA), 1 µL of *Ywhaz* probe (HEX; qRnoCIP0050810, BioRad, Hercules, CA, USA) and 1 µL of cDNA (50 ng/µL), in a total volume of 20 µL. Droplets obtained using a QX200 Droplet Generator were amplified to end-point using a T100™ Thermal Cycler. Results, acquired using a QX200™ Droplet Reader, were evaluated using QuantaSoft™ software. For final data evaluation, only data obtained from wells in which the number of droplets obtained exceeded 13,000 were used. All instruments, consumables, and reagents used in this analysis were obtained from BioRad (Hercules, CA, USA), unless otherwise stated.

### 2.6. Western Blot Analysis

Rat placentas (*n* = 5) were homogenized at 4 °C in a buffer containing 50 mM Tris-Hepes (pH 7.2), 5 mM EGTA, 5 mM EDTA, 1 mM PMSF, and 250 mM sucrose. The homogenates were filtered through gauze and centrifuged at 800× *g* for 10 min. Portions of placenta homogenates containing 80 µg total protein were mixed with loading buffer under reducing conditions [43], heated at 96 °C for 5 min and separated by SDS-PAGE on 10% polyacrylamide gels. Proteins were electrophoretically separated at 120 V and transferred to PVDF membranes (Bio-Rad). The membranes were blocked in 20 mM Tris-HCl (pH 7.6), 150 mM NaCl, 0.1% Tween 20 (TBS-T) containing 5% BSA for 1 h at room temperature and washed with TBS-T buffer.

The membranes were incubated with primary (anti-OCT3) antibody (Bioworld, BS3359, dilution 1:500) overnight at 4 °C, washed with TBS-T buffer, then incubated with specific secondary anti-rabbit horseradish peroxidase-linked antibody (Dako P0217, dilution 1:20,000), for 1 h at room temperature. Chemiluminescence signals from the membranes were enhanced using Amersham^TM^ ECL^TM^ Prime (GE Healthcare life Science), then visualized and quantified by densitometric analysis using the ChemiDoc MP Imaging system (Bio-Rad, Hercules, CA, USA). To check that membranes were equally loaded with proteins, they were probed for β-actin (Abcam, Ab8226 dilution 1:10,000) and specific secondary anti-mouse HRP antibody (Dako P0260, dilution 1:20,000).

### 2.7. Evaluation of MAO-A Activity and Lipid Peroxidation in Rat Placenta Homogenates

For monoamine oxidase A (MAO-A) activity assays, placenta homogenates (1.5–2 mg/mL; *n* = 5) were preincubated for 5 min at 37 °C, with or without phenelzine (100 µM). The reaction was initiated by adding 20 µL of 5-HT (0.5 mM) and 60 min later it was stopped by adding 40 µL of HClO_4_ (3.4 M) and placing the reaction vessels on ice for 5 min. The samples were centrifuged at 5000× *g* for 10 min, and the supernatant was used for 5-HT determination by HPLC as previously described [32]. Lipid peroxidation was estimated by measuring levels of thiobarbituric acid-reactive substances (TBARS), following published procedures [44].

### 2.8. Cells

Madin-Darby canine kidney II (MDCKII) parental cells and MDCKII cells stably transfected with P-gp (MDCKII-P-gp), BCRP (MDCKII-BCRP), and MRP2 (MDCKII-MRP2) human efflux transporters were provided by Dr. Alfred Schinkel (The Netherlands Cancer Institute). The cells were cultured in Dulbecco’s Modified Eagle’s Medium (high glucose) supplemented with 10% fetal bovine serum. All cell lines were cultured at 37 °C and 5% CO_2_; cells from passages 5–10 were used in the study.

#### In Vitro Bidirectional Transport of Selected Antidepressants

Transport assays were performed using Transwell^®^ polycarbonate membrane inserts (Cat. No. 3402; Corning, NY, USA), as previously described [45,46]. Cells were seeded at a density of 6 × 10^5^ cells/well and cultured for 4 days until confluent. At the beginning of the experiment, cells were washed with phosphate buffered saline then the transport assay was initiated by adding Opti-MEM (Thermo Fisher Scientific, Rockford, MA, USA) with [^3^H]paroxetine, [^3^H]citalopram, and [^3^H]sertraline (all at 0.18 µCi/mL concentration) to the donor compartment. Samples were collected from the acceptor compartment and their radioactivity was determined by liquid scintillation counting. Efflux ratios were calculated from measurements during the linear transport phase [45], with net efflux ratios (r_net_) normalized to efflux ratios in parental cells. The monolayers’ integrity was evaluated by measuring the transepithelial electrical resistance and FITC-dextran leakage and comparing the values to a previously published acceptable range [47] and acceptable threshold (≤1%) [45].

### 2.9. Radioisotope Analysis

The radioactivity in the experimental samples was measured by liquid scintillation counting using a Tri-Carb 2910 TR instrument (Perkin Elmer, Waltham, MA, USA).

### 2.10. Statistical Analysis

IC_50_ (half-maximal inhibitory concentration) values for the tested antidepressants’ effects on the transporters in ex vivo experiments were obtained by fitting concentration-inhibition curves through nonlinear regression analysis. Their effects on the transporters in situ were assessed by one-way ANOVA with Dunnet’s multiple comparisons test (where only the effect of antidepressant is evaluated). Two-way ANOVA with Sidak’s multiple comparisons test was used to evaluate the fetal sex-dependency of these effects (where both the effect of antidepressant drugs and fetal sex is analyzed). Effects of fetal sex on the expression of measured genes or proteins and functions were evaluated using unpaired t tests. All statistical procedures were implemented in GraphPad Prism 8.1 software (GraphPad Software, Inc., San Diego, CA, USA). Presented data are means ± SD. Asterisks in the figures indicate significance levels: * (*p* ≤ 0.05), ** (*p* ≤ 0.01), and *** (*p* ≤ 0.001).

## 3. Results

### 3.1. Concentration-Dependent Effect of Antidepressants on SERT- and OCT3-Mediated 5-HT Uptake by Human MVM and BM Vesicles

Membrane vesicles isolated from human term placenta were used to study the tested antidepressants’ inhibitory effects on 5-HT transport across the MVM and BM. Concentration-inhibition curves depicting the effects are presented in Figure 2 and results are summarized in Table 1. Overall, the antidepressants potently inhibited SERT-mediated 5-HT uptake by the MVM and OCT3-mediated uptake by the BM, with nM IC50 values. Comparison of the calculated IC_50_ values with published mean (C_a_) and maximal (C_max_) plasma concentrations in pregnant women (or concentrations in women if C_a_ and C_max_ values were not available) show that all tested antidepressants inhibited SERT and OCT3 at notably lower concentrations than their plasma C_max_.

### 3.2. Effect of Antidepressants on OCT3-Mediated 5-HT Extraction from Fetal Circulation in Perfused Rat Term Placenta

To confirm the findings from ex vivo studies, the potential capacity of all antidepressants (at 100 µM) to inhibit 5-HT uptake from fetal circulation via OCT3 was investigated in situ. Decynium-22 (10 µM) was used as a model OCT3 inhibitor. We recorded high 5-HT extraction, which was sensitive to co-administration of decynium-22, from the fetal circulation (Figure 3a). Co-administration of each of the antidepressants, except sertraline and venlafaxine, significantly decreased OCT3-mediated 5-HT uptake from the fetal circulation (Figure 3a). The antidepressants’ inhibitory potential was significantly dependent on the fetal sex (Figure 3c). More specifically, 5-HT extraction from the circulation of male fetuses was susceptible to inhibition by paroxetine, sertraline, fluoxetine, fluvoxamine, and venlafaxine. In contrast, its extraction from the circulation of female fetuses was only prone to inhibition by paroxetine, the strongest inhibitor of the tested antidepressants. The most potent inhibitors (paroxetine, citalopram and fluvoxamine) were subsequently tested at a lower, therapeutic concentration (1 µM). When fetal sex was not considered, no significant inhibition of 5-HT uptake from fetal circulation was observed (Figure 3b). However, when separating female and male placentas, paroxetine was revealed as a potent inhibitor in males also at therapeutic concentration (Figure 3d).

### 3.3. OCT3 Expression, MAO-A Activity and Lipid Peroxidation as Potential Mechanisms Accounting for Sex-Dependent Inhibition of Antidepressants on Serotonin Uptake in Rat Placenta

Several potential mechanisms were investigated to help identify the sex-dependent inhibition of 5-HT extraction observed in rat placenta. OCT3 gene and protein expression was analyzed in perfused placentas by ddPCR and western blot, respectively. However, no significant differences were observed between male and female placentas (Figure 4a,b). In addition, sex-dependent placental MAO-A activity and lipid peroxidation were determined, as a potential mechanism affecting transporter function [52]. No sex-dependent differences in MAO-A activity or lipid peroxidation were noted (Figure 4c,d).

### 3.4. Interactions of Antidepressants with Efflux Transporters In Vitro

To investigate possible interactions of antidepressants with membrane efflux transporters, citalopram, paroxetine, and sertraline (the tested antidepressants with the lowest fetal-to-maternal (F/M) concentration ratios, Table 1) were assayed in vitro using MDCKII cells stably expressing P-gp, BCRP or MRP2. Calculated r_net_ values for paroxetine and sertraline during the linear phase of transport were below 2, indicating that they had no significant effect on any of these transporters (Table 2). However, we obtained a r_net_ value for citalopram in MDCKII-P-gp cells of 2.75 ± 0.79 (Table 2), indicative of P-gp-mediated efflux of citalopram.

### 3.5. Interactions of Antidepressants with Efflux Transporters in Rat Term Placenta

Interactions of citalopram, paroxetine, and sertraline (at 1 µM) with placental efflux transporters were tested in situ using the closed-circuit perfusion setup and GF120918 (2 µM) as a model inhibitor of P-gp and BCRP. The citalopram concentration in fetal circulation did not change during the experiment, with the F/M concentration ratio at equilibrium remaining close to 1 (Figure 5c). In contrast, fetal circulation concentrations of sertraline and paroxetine steadily decreased, resulting in F/M concentration ratios at equilibrium of 0.58 and 0.61, respectively (Figure 5a,b). These results suggest active efflux of the antidepressants from the fetal to maternal circulation. Nevertheless, addition of GF 120918 (2 µM) did not affect the efflux, suggesting involvement of efflux transporters other than P-gp or BCRP.

## 4. Discussion

Prenatal use of serotonin reuptake inhibitors (SRIs) to treat maternal depression has been identified as a potential risk factor for perinatal complications such as preterm delivery, intrauterine growth restriction, pulmonary hypertension, and preeclampsia [19,53,54,55]. Moreover, epidemiological studies have associated prenatal exposure to SRIs with increased risks of autism spectrum disorders and mental diseases [56,57,58]. Previous studies have suggested that SRIs may affect 5-HT homeostasis in the fetoplacental unit, resulting in suboptimal 5-HT concentrations in utero, but have not provided direct evidence of the molecular mechanisms underlying these perturbations. Placental uptake of 5-HT is largely regulated by two distinct membrane transporters: SERT, a high-affinity and low-capacity transporter expressed in the mother-facing (MVM) membrane [59] and OCT3, a low-affinity but high-capacity transporter localized in the fetus-facing (BM) membrane [32]. Since SRIs are inhibitors of both SERT and OCT3 [38], we investigated inhibitory effects of six clinically used antidepressants on both SERT- and OCT3-mediated 5-HT uptake by the placenta. Collectively, our results clearly indicate mechanisms that may contribute to antidepressants’ poorly understood effects on the placenta-brain axis [60] during the prenatal period (Figure 6).

We first evaluated concentration-dependent effects of SRIs on 5-HT uptake by both (MVM and BM) human placental membranes. Importantly, calculated IC_50_ values were in the range of therapeutically reachable plasma concentrations (Table 1). Thus, we propose that SRIs compromise placental uptake of 5-HT not only from the maternal circulation (by inhibiting SERT) but also from fetal circulation (by inhibiting OCT3). Interestingly, we observed differences in the inhibitory patterns between the MVM and BM. High doses of SRIs could completely diminish 5-HT uptake by the MVM, while up to 30% of 5-HT uptake by the BM remained unaffected even at supra-pharmacological (100 µM) SRI concentrations. We attribute these dissimilarities to different transport capacities of SERT and OCT3 [61,62]. In addition, the BM has higher fluidity and thus higher passive permeability than the MVM, both generally [63], and specifically for 5-HT [32]. Hence, the contribution of passive, non-transporter mediated uptake of 5-HT by the BM may explain its limited antidepressant inhibition even at high concentrations (Figure 2).

To verify these ex vivo findings on organ level, we investigated the inhibitory effects of SRIs on 5-HT uptake from the fetal circulation in rat term placenta perfused in situ. This is a valuable method to investigate placental pharmacology and toxicology due to the preserved tissue integrity and sustained circulation flow. Importantly, owing to the possibility to perfuse one male and one female placenta from each dam, this technique is ideal for sex-dependent studies. On the other hand, in utero conditions for each fetus may differ in the same litter [64], and interspecies differences may also play a role since the rat placental architecture is hemochorial of labyrinthine type, while humans have hemochorial placenta of villous type [32].

At high (100 µM) concentration, all tested antidepressants significantly inhibited OCT3-mediated 5-HT uptake. We have previously reported that fetal sex influences placental uptake of 5-HT from the fetal circulation [32]. Therefore, here we considered fetal sex as a factor and found that male placentas are more sensitive to the inhibitory effect of SRIs. This may, at least partly, explain the fetal sex-dependent variation observed in studies of behavioral effects of prenatal treatment with antidepressant drugs [65] and higher risks of neurodevelopmental disorders after prenatal use of antidepressants observed in males [23,66].

It should be noted that our placenta perfusion-based experiments only revealed acute effects of SRIs on 5-HT homeostasis. Since antidepressants are prescribed chronically, and reach steady-state concentrations in both maternal and fetal circulations, further studies, including chronic administration of selected SRIs, are required to fully elucidate their long-term effects on placental homeostasis of 5-HT.

To identify mechanisms contributing to the sex-dependent differences, we assessed several aspects that could potentially affect OCT3 function. First, we investigated OCT3 expression at transcript and protein levels, as factors that could potentially contribute to transport activity. Next, since 5-HT is rapidly metabolized in the placenta by MAO-A [32], we tested the hypothesis that this activity (which inevitably affects the 5-HT concentration gradient) may differ between male and female placentas. Finally, we evaluated the level of lipid peroxidation, which may reportedly disturb membrane fluidity and permeability, thereby altering membrane transporter functions [52]. However, we detected no fetal sex-related differences in these tested processes so further studies are required to fully elucidate the variation in 5-HT uptake.

As they are lipophilic, most SRIs cross the placental barrier easily [67], but with varying F/M ratios at delivery (Table 1). Low ratios suggest limited transport of drugs across the placenta. Accordingly, current literature indicates that citalopram [68], sertraline [69], and paroxetine [68,69] are substrates of drug efflux transporters, namely P-gp, BCRP, and MRP2. Since these transporters are functionally expressed on the apical, mother-facing side of the placenta [39], we investigated their possible role in mother-to-fetus transport of paroxetine, sertraline, and citalopram. We observed active efflux of sertraline and paroxetine from the fetal to maternal circulation, resulting in F/M ratios of 0.58 and 0.61, respectively. However, lack of inhibition by GF 120918 indicates involvement of efflux mechanisms other than those mediated by P-gp or BCRP. While this extends beyond the scope of this study, we acknowledge that a thorough investigation of mother-to-fetus transport of antidepressants is necessary to fully understand placental and fetal exposure to these drugs when used in pregnancy.

All the antidepressants we tested are being used in clinical practice, including pregnant women. Of these, paroxetine is the drug most frequently associated with fetal toxicity, including increased risks of septal heart defects [70], cardiovascular malformations [21], and neonatal withdrawal symptoms [71]. Correspondingly, in this study, we identified paroxetine as the most potent inhibitor of placental uptake of 5-HT by both SERT and OCT3. We therefore suggest that strong inhibition of 5-HT transporters in the placenta and other fetal organs may, at least partly, explain the fetal toxicity of paroxetine frequently seen in clinical practice.

To the best of our knowledge, we show for the first time that antidepressants inhibit both SERT and OCT3 in the placenta. To date, interactions of antidepressants with OCT3 have only been reported in the brain [72]. We thus provide novel perspectives on the mechanisms by which antidepressants’ use during pregnancy may compromise both SERT and OCT3 functions in the placenta and subsequently dysregulate 5-HT homeostasis in the fetoplacental unit. The major strength of this study is the use of multiple placenta-based experimental models in humans and rats to address the hypotheses raised. However, our study is limited to the short-term effects of antidepressants at term; therefore, examining the outcomes of long-term drug exposure during gestation is essential for a deeper understanding of our findings. Additionally, the sex-dependent inhibitory effect observed in rats warrants further investigations to identify the mechanisms involved and to assess the relevance of these findings in humans. Lastly, future studies are required to address the potential consequences of antidepressants’ inhibition of platelet SERT activity. While fetal platelets are reportedly hypofunctional [73], antidepressant-mediated platelet SERT inhibition could affect serotonin concentration in the maternal circulation, where this mechanism is well-described [74].

## 5. Conclusions

Collectively, our findings indicate novel mechanisms whereby SRIs may affect fetoplacental homeostasis of 5-HT and contribute to poor pregnancy outcomes. Our in situ and ex vivo data indicate that SRIs block both 5-HT uptake transporters in the placenta, i.e., SERT in the mother-facing apical membrane and OCT3 in the fetus-facing basal membrane. We suggest that this effect can result in suboptimal 5-HT concentrations in the fetoplacental unit, thereby jeopardizing fetal development and/or programming.

## Figures and Tables

**Figure 1 pharmaceutics-13-01306-f001:**
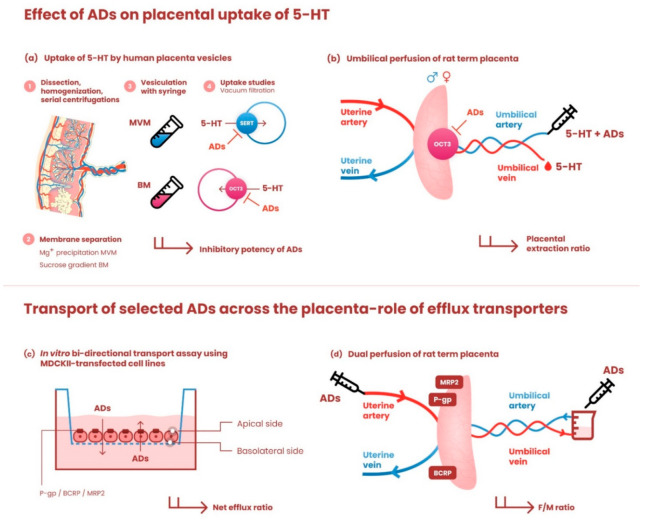
Graphical illustration of the research design and experimental models used in the study. The upper panel depicts methods employed to investigate the effects of antidepressants on placental uptake of 5-HT: (**a**) Human term placentas were dissected, homogenized, and subjected to serial centrifugation steps. Membrane separation was performed by Mg^+^ precipitation (microvillous membrane, MVM) and sucrose gradient (basal membrane, BM); subsequently, the membranes were spontaneously vesiculated by passaging through a 25-gauge needle. MVM and BM vesicles were used to evaluate the effects of antidepressants on SERT- and OCT3-mediated uptake of 5-HT, respectively. (**b**) To confirm our ex vivo findings on organ level, in situ perfused rat term placenta was employed to explore the effect of antidepressants on OCT3-mediated 5-HT uptake from the fetal circulation. The umbilical perfusion was carried out in an open-circuit setup. 5-HT with/without antidepressants was infused through the umbilical artery and the placental extraction ratio was calculated. Fetal sex was determined and considered as a potential factor. The lower panel shows methods used to investigate the role of ABC efflux transporters in transplacental passage of antidepressants: (**c**) monolayers of MDCKII cells stably transfected with P-gp, BCRP, or MRP2 were used to assess their role in transmembrane passage of [^3^H]paroxetine, [^3^H]citalopram, and [^3^H]sertraline. Net efflux ratios were calculated and normalized to efflux ratios in parental cells. (**d**) Transport of these drugs across the dually perfused rat placenta was tested in closed-circuit setup and the fetal/maternal concentration ratio at equilibrium was calculated. For detailed description of all experimental methods, see corresponding sections in Materials and Methods. ADs—antidepressants.

**Figure 2 pharmaceutics-13-01306-f002:**
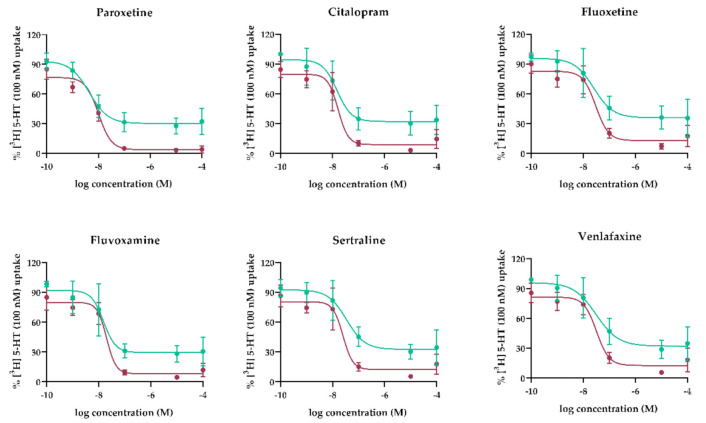
Concentration-dependent inhibition of 5-HT transport mediated by SERT (MVM, red lines) and OCT3 (BM, green lines) in membrane vesicles. Transport of [^3^H]-5-HT was measured in the presence of paroxetine, citalopram, venlafaxine, fluoxetine, fluvoxamine, and sertraline at indicated concentrations. Data shown are means ± SD (*n* ≥ 4) of percent [^3^H]5-HT uptake in the inhibitors’ presence relative to uptake in their absence. Curves were fitted by nonlinear regression analysis implemented in GraphPad Prism 8.1.

**Figure 3 pharmaceutics-13-01306-f003:**
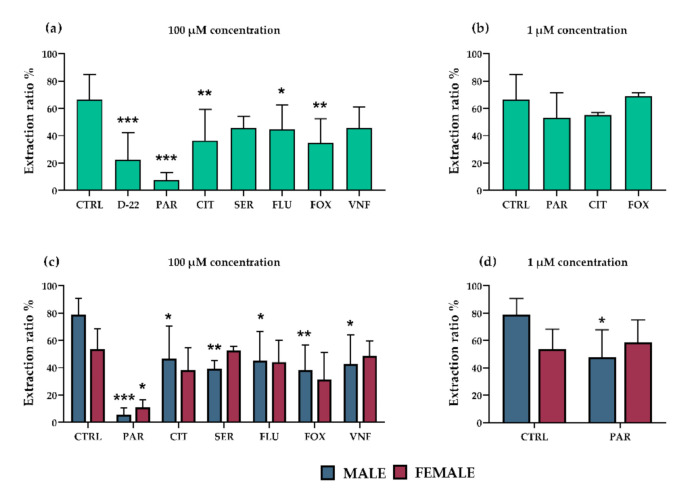
Effects of antidepressants on placental extraction of 5-HT from fetal circulation in rat placenta. Decynium 22 (10 µM) was used as a model inhibitor of OCT3. Several of the tested antidepressants significantly decreased extraction of 5-HT (relative to controls) from the fetal circulation at 100 µM concentration: most potently paroxetine, citalopram, and fluvoxamine (**a**). No significant effects of the antidepressants at 1 µM concentration were detected in analyses of samples of both fetal sexes (**b**). However, male placentas were affected more than female placentas at both tested concentrations (**c**,**d**). The values are mean ± SD; *n* = 4 dams for each condition (4 females/4 males). Differences between samples exposed to the antidepressants and unexposed controls were evaluated by one-way ANOVA with Dunnet’s multiple comparisons test (**a**,**b**). Two-way ANOVA with Sidak’s multiple comparisons test was used to evaluate the fetal sex-dependence of the antidepressants’ effects (**c**,**d**): * (*p* ≤ 0.05), ** (*p* ≤ 0.01), *** (*p* ≤ 0.001). Abbreviations: CTRL—Controls, D-22—decynium 22, PAR—paroxetine, CIT—citalopram, SER—sertraline, FLU—fluoxetine, FOX—fluvoxamine, VNF—venlafaxine.

**Figure 4 pharmaceutics-13-01306-f004:**
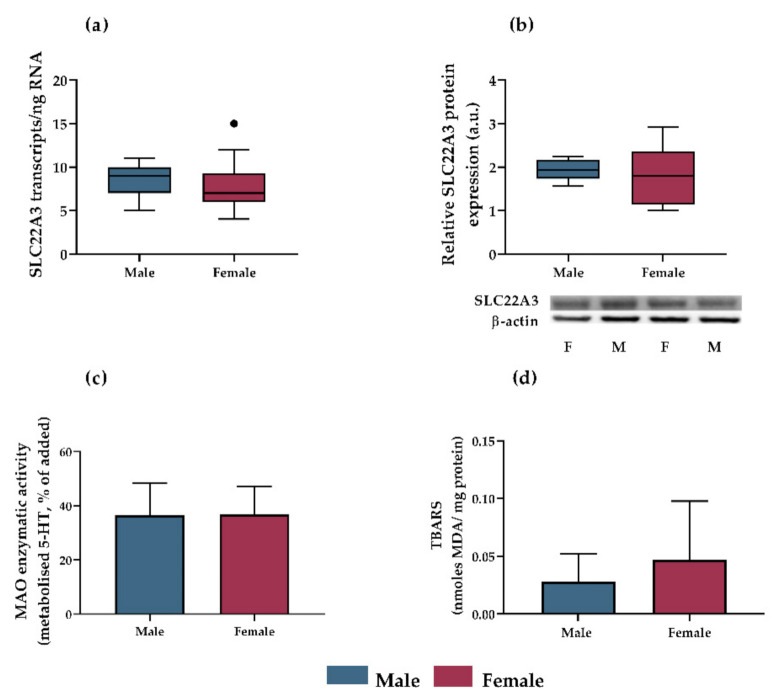
Evaluation of OCT3 expression, MAO-A activity, and lipid peroxidation as potential mechanisms accounting for sex-dependent inhibition of 5-HT extraction by rat placenta. OCT3 gene expression ((**a**), assessed by ddPCR; presented as Tukey plots with whiskers extending to 1.5 times IQR, outliers are shown as individual points; *n* ≥ 18) and protein expression ((**b**), assessed by western blot analysis; presented as Tukey plots with whiskers extending to 1.5 times IQR; *n* = 5) were independent of fetal sex (evaluated by unpaired t test). Protein expression was normalized to β-actin as a loading control; representative immunoblots for target proteins and β-actin are also shown. No differences in MAO-A activity ((**c**), presented as means ± SD; *n* = 5) or lipid peroxidation ((**d**), presented as means ± SD; *n* = 5) between male and female placentas were detected.

**Figure 5 pharmaceutics-13-01306-f005:**
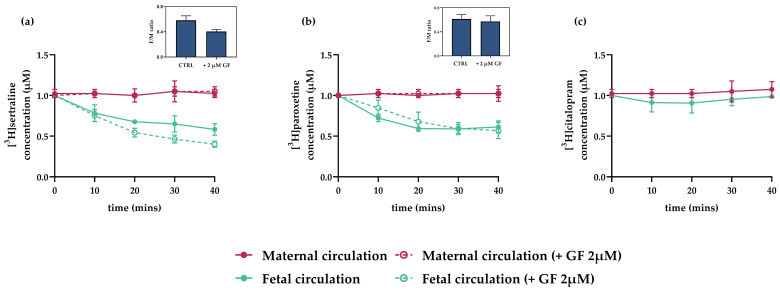
Interactions of antidepressants with placental efflux transporters in dually perfused rat term placenta in closed experimental setup. Reductions in [^3^H]sertraline and [^3^H]paroxetine concentrations in the fetal circulation were observed, suggesting active efflux to the maternal circulation (**a**,**b**). However, these reductions were not inhibited by addition of GF 120918 (2 µM), a model P-gp and BCRP inhibitor. [^3^H]citalopram concentrations were stable throughout the perfusion experiment (**c**) indicative of passive transport across the placenta. Insets show ratios of concentrations of tested drugs in fetal and maternal circulations (F/M ratios) in the absence and presence of GF 120918 (2 µM), at the final sampling point (40 min). Data are means ± SD; *n* > 3.

**Figure 6 pharmaceutics-13-01306-f006:**
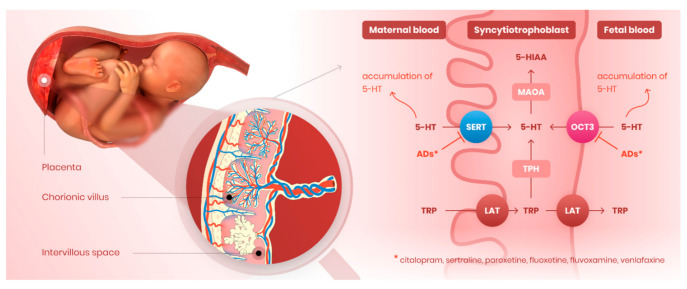
Novel perspectives of antidepressants’ effect on serotonin homeostasis in the placenta. Serotonin reuptake inhibitors tested in our study (citalopram, sertraline, paroxetine, fluoxetine, fluvoxamine, venlafaxine) block the activity of both SERT and OCT3 transporters localized on the microvillous and basal membrane, respectively. Antidepressants thus compromise placental clearance of serotonin from maternal and fetal circulations, resulting in inadequate serotonin concentrations in the fetoplacental unit. By this effect, these drugs may disrupt proper placental and/or fetal development and programming. ADs—antidepressants.

**Table 1 pharmaceutics-13-01306-t001:** Heat-map of average antidepressant drug concentrations resulting in 50% inhibition (IC_50_) of 5-HT uptake by MVM (SERT-mediated) and BM (OCT3-mediated). 5-HT transport inhibition clearly occurs at clinically relevant concentrations, according to published C_max_ [48,49], C_a_ [49,50] and F/M ratio [51] values. The red color intensity indicates the inhibitory potency of antidepressants based on the obtained IC_50_ values.

	Antidepressant	nM					F/M Ratio
		IC_50_	IC_50_	C_a_	C_max_	C_a;fetal_	
		MVM	BM				
Stronger inhibitor	Paroxetine	10.09	4.39	455	1000	209	0.46
	Citalopram	16.52	15.07	216	1200	147	0.68
	Fluvoxamine	19.95	15.21	446	1300	348	0.78
Weaker inhibitor	Fluoxetine	29.99	25.18	557	1900	368	0.66
	Sertraline	25.18	34.43	104	1300	53	0.51
	Venlafaxine	32.07	30.76	225	1400	162	0.72

Abbreviations: Cmax—maximum steady-state plasma concentration; Ca—average maternal plasma concentration; Ca;fetal—average fetal plasma concentration; F/M ratio—antidepressant’s fetal-to-maternal concentration ratio.

**Table 2 pharmaceutics-13-01306-t002:** Net efflux ratios (r_net_) of [^3^H]citalopram, [^3^H]paroxetine and [^3^H]sertraline across MDCKII-P-gp, MDCKII-BCRP and MDCKII-MRP2 monolayers. Data shown are means *±* SD; *n* = 3.

Efflux Transporter	Citalopram	Paroxetine	Sertraline
MDCKII-P-gp	2.75 ± 0.79	1.39 ± 0.30	1.17 ± 0.20
MDCKII-BCRP	1.18 ± 0.32	0.92 ± 0.02	1.11 ± 0.61
MDCKII-MRP2	0.61 ± 0.11	-	1.69 ± 0.36

Abbreviation: r_net_—efflux ratios obtained for P-gp-, BCRP- and MRP2-transfected MDCKII cells, normalized to the ratio obtained for the parental cells.

## Data Availability

Not applicable.

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
