# Peer review of "Effect of Selected Antidepressants on Placental Homeostasis of Serotonin: Maternal and Fetal Perspectives"

_pharmaceutics, 2021, doi:10.3390/pharmaceutics13081306_

Round 1

Reviewer 1 Report

In this study the authors deal about effect of selected antidepressants on placental homeostasis of 2 serotonin: maternal and fetal perspectives

In order to increase the value of the manuscript, few things need to be addressed:

1.Introduction section: I suggest adding more data about 5-HT and fetal brain development, Prenatal exposure to maternal depression and antidepressants, effects on fetal brain development and long-term consequences

2. The MS is difficult to follow, a flowchart with the research’s design would be welcomed for readers.

3.Line 278: Table 1 Legend: It is necessary to explain the red colors in detail and their meaning.

4. I suggest to extend the Discussion section, which should contain at least the following aspects: limitations and strengths of this study, the novelty of the study, comparisons with more similar studies in the literature.

5. What clinical therapeutic perspectives for humans do this study have

6. Also, the authors must add a schematic diagram of potential working mechanisms.

The paper looks informative and would be useful for the research community.

 Consider revision accordingly.

Author Response

We would like to thank both reviewers for their careful reading of our manuscript and their stimulating comments and suggestions. We have tried our best to address the issues raised and we believe it has improved our manuscript significantly. For step-by-step replies to all questions, see below.

Reviewer #1

In this study the authors deal about effect of selected antidepressants on placental homeostasis of 2 serotonin: maternal and fetal perspectives

In order to increase the value of the manuscript, few things need to be addressed:

1.Introduction section: I suggest adding more data about 5-HT and fetal brain development, Prenatal exposure to maternal depression and antidepressants, effects on fetal brain development and long-term consequences

Thank you for your comment. We have added more information in the introduction section; see yellow highlighted text in the Introduction part of our revised manuscript.

  1. The MS is difficult to follow, a flowchart with the research’s design would be welcomed for readers.

Thank you for this very helpful suggestion! We agree with your point and we have generated a new figure (see Figure 1 in the revised manuscript) that graphically illustrates our research design and experimental approaches used in our study. In two panels, it highlights the two points of our study, specifically: i) the inhibitory effect of antidepressants on placental serotonin uptake by SERT and OCT3 (tested ex vivo and in situ), and ii) interaction of selected antidepressants with efflux transporters (tested in vitro and in situ).

3.Line 278: Table 1 Legend: It is necessary to explain the red colors in detail and their meaning.

We have amended the table legend accordingly. It now reads: “The red color intensity indicates the inhibitory potency of antidepressants based on the obtained IC50 values”.

  1. I suggest to extend the Discussion section, which should contain at least the following aspects: limitations and strengths of this study, the novelty of the study, comparisons with more similar studies in the literature.

Thank you for this suggestion. We agree and thus we have added a new paragraph in the Discussion section highlighting the strengths and limitations of our study and its novelty in relation to the current state of knowledge. It now reads:

“To the best of our knowledge, we show for the first time that antidepressants inhibit both SERT and OCT3 in the placenta. To date, interactions of antidepressants with OCT3 have only been reported in the brain [72]. We thus provide novel perspectives on the mechanisms by which antidepressants’ use during pregnancy may compromise both SERT and OCT3 functions in the placenta and subsequently dysregulate 5-HT homeostasis in the fetoplacental unit. The major strength of this study is the use of multiple placenta-based experimental models in humans and rats to address the hypotheses raised. However, our study is limited to the short-term effects of antidepressants at term; therefore, examining the outcomes of long-term drug exposure during gestation is essential for a deeper understanding of our findings. Additionally, the sex-dependent inhibitory effect observed in rats warrants further investigations to identify the mechanisms involved and to assess the relevance of these findings in humans. Lastly, future studies are required to address the potential consequences of antidepressants’ inhibition of platelet SERT activity. While fetal platelets are reportedly hypofunctional [73], antidepressant-mediated platelet SERT inhibition could affect serotonin concentration in the maternal circulation, where this mechanism is well-described [74]”.

  1. What clinical therapeutic perspectives for humans do this study have

The main aim of our study was to decipher, at least partly, possible mechanism of poor pregnancy outcomes upon prenatal use of antidepressants repeatedly reported in epidemiological studies. We confirmed our hypothezis that antidepressants affect placental serotonin homeostasis by inhibiting serotonin transporters on both sides of the placenta. We admit that the clinical therapeutic perspective for humans requires further investigation and is beyond the scope of this study. This information has been added to the last paragraph in Discussion as mentioned above (question 4).

  1. Also, the authors must add a schematic diagram of potential working mechanisms.

In figure 6 we do our best to summarize the results of our study and show antidepressants’ inhibitory mechanisms identified. We believe that presenting only the mechanisms investigated and confirmed in our study is appropriate; we are afraid that the depiction of other alternative working mechanisms, not investigated here, would be misleading for the reader. 

The paper looks informative and would be useful for the research community.

We really appreciate your supportive comment!

Reviewer 2 Report

Manuscript Pharmaceutics "Effect of selected antidepressants on placental homeostasis of 2 serotonin: maternal and fetal perspectives" By H. Horackova et al.

In this manuscript, the authors are interested in serotonin selective reuptake inhibitor (SSRI) antidepressant effects on placenta, at late stage of gestation. The authors hypothesize that antidepressants act by inhibiting serotonin transporter (SERT) and organic cation transporter 3 (OCT3) in the maternal- and fetal-facing placental membranes, respectively. They report that paroxetine, citalopram, fluoxetine, fluvoxamine, sertraline and venlafaxine inhibited SERT- and OCT3-mediated serotonin uptake in a dose-dependent manner. They found that SSRIs completely blocked 5-HT uptake by the MVM, while most 5-HT uptake by the BM remained unaffected even at high dose, in agreement with different affinity for SERT and OCT3. They found that the effect of antidepressants on rat placenta was sex dependent. Then, they further showed that the placental efflux transporters did not compromise mother-to-fetus transport of antidepressants. The authors conclude that SSRIs have the potential to affect serotonin levels in the placenta or fetus, potentially affecting fetal development.

Indeed, understanding how placenta is controlling the proper serotonin amount both at the maternal and fetal sides is an important issue. This small set of data is sound and provides some putative answers to this question. It remains, however, unclear why these effects are more affecting male placenta. Furthermore, one issue that is not discussed is the presence on both maternal and fetal circulation of platelets that express the same SERT and that clearly participate in 5-HT reuptake. Finally, it remains to be understood how is operated the mother-to-fetus transport of antidepressants.

Author Response

We would like to thank both reviewers for their careful reading of our manuscript and their stimulating comments and suggestions. We have tried our best to address the issues raised and we believe it has improved our manuscript significantly. For step-by-step replies to all questions, see below.

Reviewer #2

Manuscript Pharmaceutics "Effect of selected antidepressants on placental homeostasis of 2 serotonin: maternal and fetal perspectives" By H. Horackova et al.

In this manuscript, the authors are interested in serotonin selective reuptake inhibitor (SSRI) antidepressant effects on placenta, at late stage of gestation. The authors hypothesize that antidepressants act by inhibiting serotonin transporter (SERT) and organic cation transporter 3 (OCT3) in the maternal- and fetal-facing placental membranes, respectively. They report that paroxetine, citalopram, fluoxetine, fluvoxamine, sertraline and venlafaxine inhibited SERT- and OCT3-mediated serotonin uptake in a dose-dependent manner. They found that SSRIs completely blocked 5-HT uptake by the MVM, while most 5-HT uptake by the BM remained unaffected even at high dose, in agreement with different affinity for SERT and OCT3. They found that the effect of antidepressants on rat placenta was sex dependent. Then, they further showed that the placental efflux transporters did not compromise mother-to-fetus transport of antidepressants. The authors conclude that SSRIs have the potential to affect serotonin levels in the placenta or fetus, potentially affecting fetal development.

Indeed, understanding how placenta is controlling the proper serotonin amount both at the maternal and fetal sides is an important issue. This small set of data is sound and provides some putative answers to this question. It remains, however, unclear why these effects are more affecting male placenta.

While we tried our best to address potential mechanisms involved, we agree that the sex-dependent inhibitory effect observed in our in situ studies requires further investigations to identify the mechanism(s) involved. To address this point, we have included a sentence in the newly added paragraph of the Discussion section focusing on study strengths and limitations, which reads: “Additionally, the sex-dependent inhibitory effect observed in rats warrants further investigations to identify the mechanisms involved and to assess the relevance of these findings in humans”.

Furthermore, one issue that is not discussed is the presence on both maternal and fetal circulation of platelets that express the same SERT and that clearly participate in 5-HT reuptake.

Regarding the presence of platelets in the maternal and fetal circulation, limited evidence exists whether fetal serotonin is contained in platelets, as is the case in adults. While it has been reported that uptake of serotonin by platelets in the fetal circulation is similar to maternal one (PMID: 1247495), more recent studies show lower platelet number and hypofunction in the developing fetus (PMID: 28428216) as well as low platelet serotonin count at birth (PMID: 2253357 and 15240861). Nonetheless, we agree this is a very important aspect to consider; therefore, we have discussed this issue in the revised version of the manuscript (last paragraph of Discussion): “Lastly, future studies are required to address the potential consequences of antidepressants inhibition of platelet SERT activity. While fetal platelets are reportedly hypofunctional [73], antidepressant-mediated platelet SERT inhibition could affect serotonin concentration in the maternal circulation, where this mechanism is well-described [74].”

Finally, it remains to be understood how is operated the mother-to-fetus transport of antidepressants.

We agree that a thorough investigation of mother-to-fetus transport of antidepressants is necessary to fully understand placental and fetal exposure to these drugs when used in pregnancy. However, this issue is beyond the scope of our paper; nevertheless, in the revised version of the manuscript, we acknowledge that this mechanism requires investigation in future studies: “While this extends beyond the scope of this study, we acknowledge that a thorough investigation of mother-to-fetus transport of antidepressants is necessary to fully understand placental and fetal exposure to these drugs when used in pregnancy."

Round 2

Reviewer 1 Report

No answer given.